# Macrophage PPARγ inhibits Gpr132 to mediate the anti-tumor effects of rosiglitazone

Wing Yin Cheng[1], HoangDinh Huynh[1], Peiwen Chen[1], Samuel Peña-Llopis[1], Yihong Wan[1,2]*

[1]Department of Pharmacology, The University of Texas Southwestern Medical Center, Dallas, United States; [2]Simmons Cancer Center, The University of Texas Southwestern Medical Center, Dallas, United States

**Abstract** Tumor-associated macrophage (TAM) significantly contributes to cancer progression. Human cancer is enhanced by PPARγ loss-of-function mutations, but inhibited by PPARγ agonists such as TZD diabetes drugs including rosiglitazone. However, it remains enigmatic whether and how macrophage contributes to PPARγ tumor-suppressive functions. Here we report that macrophage PPARγ deletion in mice not only exacerbates mammary tumor development but also impairs the anti-tumor effects of rosiglitazone. Mechanistically, we identify Gpr132 as a novel direct PPARγ target in macrophage whose expression is enhanced by PPARγ loss but repressed by PPARγ activation. Functionally, macrophage Gpr132 is pro-inflammatory and pro-tumor. Genetic Gpr132 deletion not only retards inflammation and cancer growth but also abrogates the anti-tumor effects of PPARγ and rosiglitazone. Pharmacological Gpr132 inhibition significantly impedes mammary tumor malignancy. These findings uncover macrophage PPARγ and Gpr132 as critical TAM modulators, new cancer therapeutic targets, and essential mediators of TZD anti-cancer effects.

*For correspondence: yihong.wan@utsouthwestern.edu

**Competing interests:** The authors declare that no competing interests exist.

## Introduction

How immune cells in the tumor microenvironment modulate cancer malignancy is a fundamental and fascinating question with tremendous therapeutic significance for cancer intervention. Emerging evidence supports a functional association between inflammation and cancer. Chronic inflammation is implicated in >15% of cancers (*Coussens and Werb, 2002*) and shown to promote tumorigenesis (*Blaser et al., 1995*; *Kuper et al., 2000*; *Scholl et al., 1994*; *Shacter and Weitzman, 2002*). As a key player in inflammation and cancer progression, TAM strongly correlates with poor cancer prognosis (*Bingle et al., 2002*; *Noy and Pollard, 2014*; *Qian and Pollard, 2010*; *Ruffell et al., 2012*). For example, overexpression of macrophage colony-stimulating factor 1 (CSF1 or M-CSF) leads to accelerated tumor progression in mice and human (*Lin et al., 2001*; *Scholl et al., 1994*). Moreover, TAM also modulates therapeutic responses (*Ruffell and Coussens, 2015*). Although numerous clinical studies and experimental mouse models support that macrophages generally play a pro-cancer role, anti-tumor property has also been reported for certain subtypes of macrophages, suggesting that macrophage regulation of cancer malignancy is pleotropic and context-dependent (*Krzeszinski and Wan, 2015*; *Noy and Pollard, 2014*; *Qian and Pollard, 2010*; *Ruffell et al., 2012*; *Ruffell and Coussens, 2015*).

Peroxisome proliferator-activated receptor gamma (PPARγ) is a nuclear receptor and a transcription factor that regulates a myriad of physiological processes (*Ahmadian et al., 2013*; *Lefterova et al., 2014*). PPARγ loss-of-function mutations have been associated with human cancer development (*Aldred et al., 2003*; *Sarraf et al., 1999*). Synthetic PPARγ agonists, such as the TZD

**eLife digest** The immune system can both contribute to cancer progression and restrain the growth of tumors. Some immune cells – called macrophages – create an inflammatory environment around a tumor, which can support the spread of the cancer cells.

Independent observations and experiments have shown that a protein called PPARγ can suppress the development and growth of tumors. Drugs called thiazolidinediones (or TZDs for short), which are normally used to treat type 2 diabetes, activate PPARγ and therefore have anti-tumor effects. However, it is not fully understood how PPARγ and TZDs suppress tumor development.

Cheng et al. hypothesized that the PPARγ protein and TZDs can inhibit the activity of the inflammatory macrophages that help tumors to develop. To test this, mice were genetically engineered so that their macrophages could not produce the PPARγ protein. These engineered mice were more likely to develop breast cancer than normal. Furthermore, the breast tumors in the modified mice did not shrink when they were treated with TZDs, whereas the tumors of normal mice did.

Cheng et al. also found that PPARγ inhibits the ability of macrophages to produce a protein called Gpr132, which itself contributes to inflammation and allows breast cancer cells to grow. Mice that were unable to produce Grp132 displayed less inflammation, and cancer growth was blocked. Drugs that inhibited the activity of Grp132 in normal mice also reduced the ability of breast tumors to spread.

Future experiments will need to examine exactly how the Gpr132 proteins produced by macrophages communicate with the cancer cells. Furthermore, developing new drugs that can inhibit Gpr132 could ultimately lead to more effective treatments for cancer.

anti-diabetic drugs rosiglitazone and pioglitazone, have been implicated to inhibit tumor malignancy (*Apostoli et al., 2015*; *Bosetti et al., 2013*; *Drzewoski et al., 2011*; *Feng et al., 2011*; *Fenner and Elstner, 2005*; *Fröhlich and Wahl, 2015*; *Kumar et al., 2009*; *Monami et al., 2014*; *Skelhorne-Gross et al., 2012*; *Uray et al., 2012*). These evidences suggest that PPARγ exerts anti-tumor effects, although a lack of correlation or pro-tumor effects have also been reported (*Saez et al., 2004*, *1998*). Limited clinical trials to date are inconclusive on the effects of TZDs on human cancer outcome (*Burstein et al., 2003*; *Mueller et al., 2000*). Provocatively, a meta-analysis of randomized clinical trials reveal that the incidence of cancer malignancies was significantly lower in rosiglitazone-treated patients than in control groups, although rosiglitazone did not significantly modify the risk of cancer (*Monami et al., 2008*). These findings not only support an anti-tumor role of TZDs but also suggest that TZDs may act to impede tumor progression rather than tumor initiation.

Importantly, epidemiological studies suggest a bidirectional association between diabetes and cancer: diabetes (especially type 2) correlates with higher risk of cancers including breast cancer (*Park et al., 2014*; *Smith and Gale, 2009*, *2010*); conversely, 8–18% of newly diagnosed cancer patients are diabetic, and cancer patients with preexisting diabetes are 50% more likely to die after surgery (*Barone et al., 2010*; *Richardson and Pollack, 2005*). Therefore, it is of paramount importance to understand how anti-diabetic drugs influence cancer for a better treatment of both cancer and diabetes.

Previous studies largely focused on the direct effects of PPARγ on cancer cells. TZDs were shown to promote terminal differentiation, reduce proliferation and trigger lipid accumulation in human breast cancer cells and liposarcoma cells (*Mueller et al., 1998*; *Tontonoz et al., 1997*). More recently, the anti-proliferative effect of pioglitazone was reported to involve a metabolic switch in lung and breast cancer cells (*Srivastava et al., 2014*). However, whether and how PPARγ in macrophages modulates cancer progression is unknown. Moreover, previous studies heavily relied on the usage of PPARγ ligands, which may exert PPARγ-independent and/or physiologically irrelevant effects; whereas in vivo genetic dissection of the specific PPARγ functions in each cell type in the cancer milieu is lacking.

We previously reported that female mice with PPARγ deletion in the hematopoietic and endothelial cells developed inflammation in their lactating mammary gland. This led to the production of

inflammatory milk, which triggered systemic inflammation in the nursing neonates manifested as a transient fur loss (*Wan et al., 2007b*). These intriguing observations suggest that PPARγ plays an anti-inflammatory role in macrophage and mammary gland, which may influence breast cancer. We hypothesize that PPARγ in macrophages impedes breast cancer development by inhibiting inflammation. Using a series of genetic and pharmacological, gain- and loss-of-function, in vitro and in vivo approaches, here we uncover macrophage PPARγ as an important suppressor of breast cancer progression and a key mediator of the anti-tumor effects of rosiglitazone that functions by repressing a novel target Gpr132 in macrophages.

## Results

### Macrophage PPARγ deletion enhances tumor growth in vivo

We generated macrophage PPARγ knockout mice (mf-g-KO) by breeding PPARγ flox mice with Tie2-Cre or Lysozyme-Cre (LyzCre). Tie2Cre deleted PPARγ in hematopoietic cells and endothelial cells as we previously described (*Wan et al., 2007a*, *2007b*). LyzCre deleted PPARγ in the myeloid lineage (*Clausen et al., 1999*). These two mf-g-KO models are complementary with different pros and cons: although Lyz-g-KO mice permit a more specific macrophage PPARγ deletion, Tie2-g-KO mice confer a more complete macrophage PPARγ deletion (89%) compared with Lyz-g-KO mice (79%). Thus, we compared $Pparg^{flox/flox};Cre^{tg/+}$ KO mice with $Pparg^{flox/flox};Cre^{+/+}$littermate controls using both models.

To determine the effects of macrophage PPARγ deletion on breast cancer progression, we performed mammary fat pad orthotopic injections of C57BL/6J-compatible mouse breast cancer cells EO771 in female mice, and followed tumor growth by measuring tumor size. Compared to the littermate controls, both Tie2Cre-induced and LyzCre-induced mf-g-KO mice showed enhanced tumor development as indicated by earlier onset and larger tumor volume (*Figure 1A–B*). These results indicate that the pro-tumor effect observed was largely caused by PPARγ deletion in myeloid cells such as macrophages. Staining for Ki67 and phospho histone H3 (PH3) in the tumor sections showed increased cell proliferation in mf-g-KO mice (*Figure 1C–D*). To assess a different cancer model, we obtained another C57BL/6-compatible mouse cell line (Py230) that was derived from spontaneous mammary tumors in C57BL/6 MMTV-PyMT female transgenic mice. Py230 cell injection into the mammary fat pad showed that tumor growth was also exacerbated in mf-g-KO mice compared with control mice (*Figure 1—figure supplement 1A*). These findings suggest that macrophage PPARγ inhibits tumor growth in vivo.

### Macrophage PPARγ deletion increased TAM abundance in vivo

We collected tumor tissues, bone marrow cells and spleen cells from tumor-bearing mf-g-KO or control mice, and compared gene expression. The results showed that the expression of pro-inflammatory genes was increased in these PPARγ-deficient cells and tissues, including *COX-2*, *MMP9*, *MCP-1*, *TNFα* and *IL-1β* (*Figure 1E–G*) (*Figure 1—figure supplement 1B*). In contrast, the expression of M2 macrophage markers such as Arginase 1 was decreased (*Figure 1—figure supplement 1B*). These observations were consistent with previous reports from several laboratories including our own group that PPARγ deficiency promotes inflammatory macrophage activation but attenuates M2 phenotype (*Odegaard et al., 2007*; *Ricote et al., 1998*; *Straus and Glass, 2007*; *Wan et al., 2007b*).

Macrophage infiltration into tumors is a strong indicator for cancer malignancy and poor prognosis (*Komohara et al., 2014*; *Ruffell and Coussens, 2015*; *Zhang et al., 2012*). Immunofluorescence staining using CD11b and F4/80 markers revealed enhanced TAM recruitment in both Tie2-g-KO and Lyz-g-KO mice compared with control mice (*Figure 1H*) (*Figure 1—figure supplement 1C–D*). This is in line with previous findings that PPARγ-deficient macrophages exhibit increased migration and CCR2 expression (*Babaev et al., 2005*), whereas TZD treatment suppresses macrophage migration and CCR2 expression (*Barlic et al., 2006*; *Chen et al., 2005*; *Han et al., 2000*; *Shah et al., 2007*).

Consistent with the reports that PPARγ agonists inhibit angiogenesis (*Goetze et al., 2002*; *Keshamouni et al., 2005*; *Scoditti et al., 2010*), we found that the number of blood vessels in tumor sections was increased in Tie2-g-KO mice but unaltered in Lyz-g-KO mice (*Figure 1—figure*

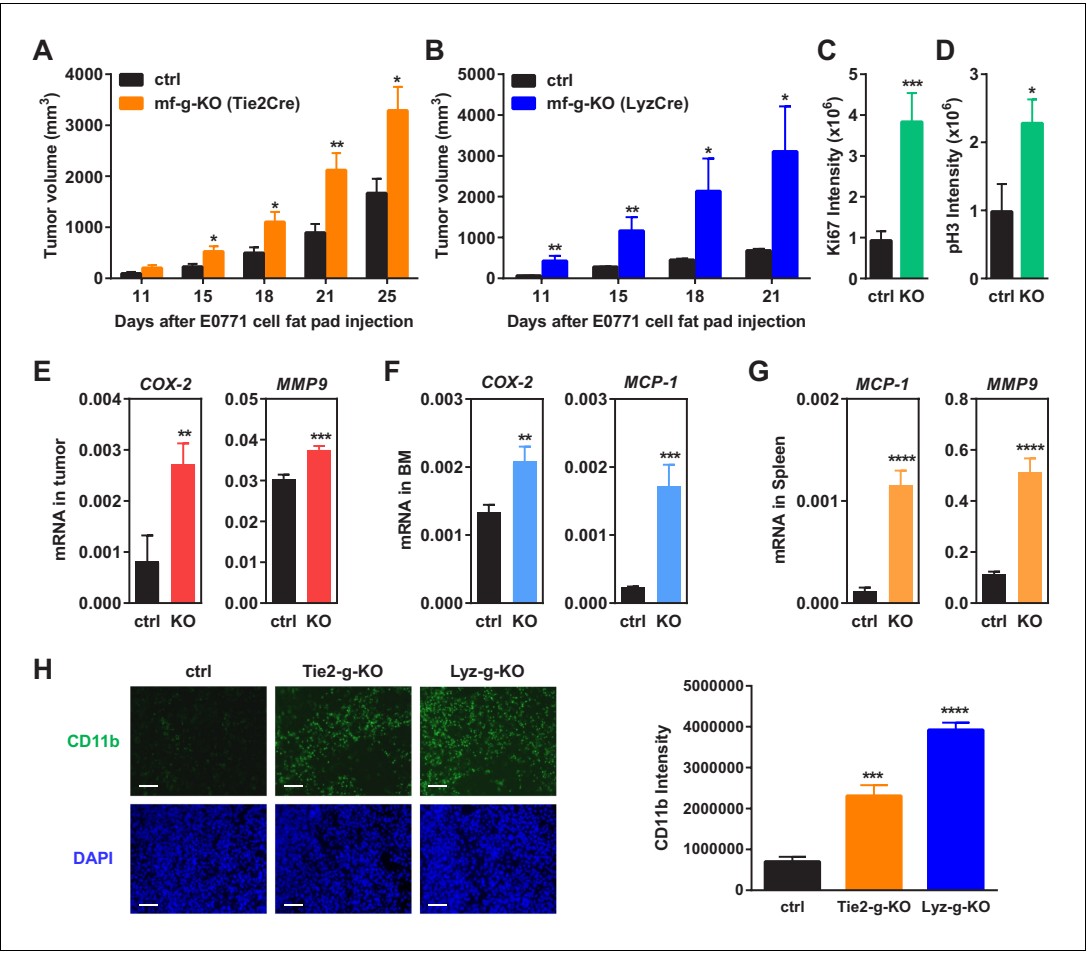

**Figure 1.** Macrophage PPARγ deletion enhances mammary tumor growth in vivo. (**A**) Tie2Cre-induced mf-g-KO mice (n = 26) showed enhanced tumor growth compared to control mice (n = 16) as indicated by earlier onset and larger tumor volume. EO771 mouse mammary tumor cells were injected into the mammary fat pad of 6–8 weeks old female mice. (**B**) LyzCre-induced mf-g-KO mice (n = 6) showed augmented tumor growth compared to control mice (n = 6) as indicated by earlier onset and larger tumor volume. (**C–D**) Quantification of cell proliferation markers Ki67 (**C**) and phosphor histone H3 (PH3) (**D**) in tumor sections showed increased cell proliferation in mf-g-KO mice (n = 4). (**E–G**) RT-QPCR analyses showed an increased expression of pro-inflammatory genes in tumor tissues (**E**), bone marrow (BM) (**F**) and spleen (**G**) from mf-g-KO mice (n = 3). (**H**) Immunofluorescence staining of tumor sections for macrophage marker CD11b showed an enhanced macrophage recruitment in the tumors from both Tie2Cre- and LyzCre-induced mf-g-KO mice compared with control mice (n = 4). All tumors were collected 21 days after cancer cell injection. Error bars, SD; *p<0.05; **p<0.01; ***p<0.005; ****p<0.001; n.s. non-significant.

The following figure supplement is available for figure 1:

**Figure supplement 1.** Additional analyses of tumors.

---

supplement 1E–F), further indicating that PPARγ deficiency in macrophage alone is sufficient to augment tumor growth independent of changes in angiogenesis. Together, these findings suggest that macrophage PPARγ deletion changes both the number and property of TAMs to establish a pro-inflammatory tumor environment.

## PPARγ-deficient macrophages promote cancer cell proliferation in vitro

To determine if PPARγ-deficient macrophages regulate cancer cell behavior in the absence of other components in the tumor microenvironment such as fibroblasts and extracellular matrix, we performed macrophage and cancer cell co-culture experiments in vitro (**Figure 2A**). Mouse

macrophages were differentiated from the progenitors in bone marrow or spleen and then co-cultured with a luciferase-labelled subline of the MDA-MB-231 human breast cancer cell line (1833 cells). Specific quantification of tumor cell proliferation was achieved by the luciferase output as only the cancer cells, but not the macrophages, were tagged with a luciferase reporter. The results showed that tumor cell proliferation was significantly augmented by PPARγ-deficient macrophages compared with WT control macrophages (*Figure 2B*). Consistent with this observation, co-culture with PPARγ-deficient macrophages also led to an increased tumor cell colony formation (*Figure 2C*). Since mouse macrophages and human cancer cells were from different species, mRNA expression in these two cell types in the co-culture setting could be distinguished by species-specific QPCR primers. We found that co-culture with PPARγ-deficient macrophages resulted in higher expression of proliferation markers and lower expression of apoptosis markers in cancer cells compared with WT control macrophages (*Figure 2D–E*).

In accordance with our in vivo observations (*Figure 1*), PPARγ-deficient macrophages exhibited elevated expression of pro-inflammatory genes such as *COX-2, MCP-1* and *MMP-9* but decreased M2 macrophage markers such as Arginase-1 (*Figure 2F*) (*Figure 2—figure supplement 1A*). In addition, PPARγ-deficient macrophages displayed higher levels of anti-apoptotic genes and lower levels of pro-apoptotic genes (*Figure 2G*), indicating an augmented survival. Moreover, PPARγ-deficient macrophages showed increased proliferation, measured by ATP content (*Figure 2H*) or MTT assay (not shown). Our in vitro findings further support our in vivo observations that the increased number and pro-inflammatory property of PPARγ-deficient macrophages are sufficient to promote tumor progression.

## Rosiglitazone activation of macrophage PPARγ inhibits cancer cell proliferation in vitro

As a complementary approach to our loss-of-function genetic approach, we next performed gain-of-function pharmacological experiment to assess the effect of rosiglitazone activation of macrophage PPARγ on cancer cells. Mouse macrophages were pre-treated with rosiglitazone or vehicle control; rosiglitazone was removed by the medium change before human cancer cells were seeded for co-culture (*Figure 2A*). The results showed that cancer cell growth was significantly inhibited when co-cultured with rosiglitazone-treated WT macrophages compared with vehicle-treated WT macrophages (*Figure 2I*). Importantly, this rosiglitazone effect was macrophage-PPARγ-dependent because tumor cell proliferation was increased equally when co-cultured with PPARγ-deficient macrophages regardless of rosiglitazone or vehicle treatment (*Figure 2I*). As a positive control, rosiglitazone induction of a previously reported PPARγ target gene LXRα (*Chawla et al., 2001*) was observed in WT macrophages but not g-KO macrophages (*Figure 2—figure supplement 1B*). Together, these findings indicate that activation of macrophage PPARγ by either endogenous or synthetic agonists suppresses tumor growth.

## Macrophage PPARγ is a key mediator of the anti-tumor effect of rosiglitazone in vivo

To assess the functional significance of macrophage PPARγ in the pharmacological effects of rosiglitazone, we treated mf-g-KO mice and littermate controls with rosiglitazone or vehicle control starting four days after cancer cell injection. The results show that the ability of rosiglitazone to suppress breast cancer growth was significantly attenuated in mf-g-KO mice (*Figure 2J*). Consistent with this observation, the ability of rosiglitazone to reduce tumor-associated macrophages was also impaired in mf-g-KO mice (*Figure 2—figure supplement 1C*). This indicates that the macrophage is an essential cell type that is required for the anti-tumor function of rosiglitazone.

## Macrophage PPARγ represses Gpr132 expression

To understand how PPARγ alters the transcription program in macrophages to control cancer cell proliferation, we next set out to identify the key PPARγ target genes. Our experiments reveal that tumor cell proliferation could be significantly enhanced by co-culture with PPARγ-deficient macrophages but not by the conditioned medium from PPARγ-deficient macrophages (*Figure 3A–B*), indicating that physical contact between macrophages and cancer cells may be required and thus the key tumor-modulating PPARγ target gene in macrophages likely encodes a membrane protein. By

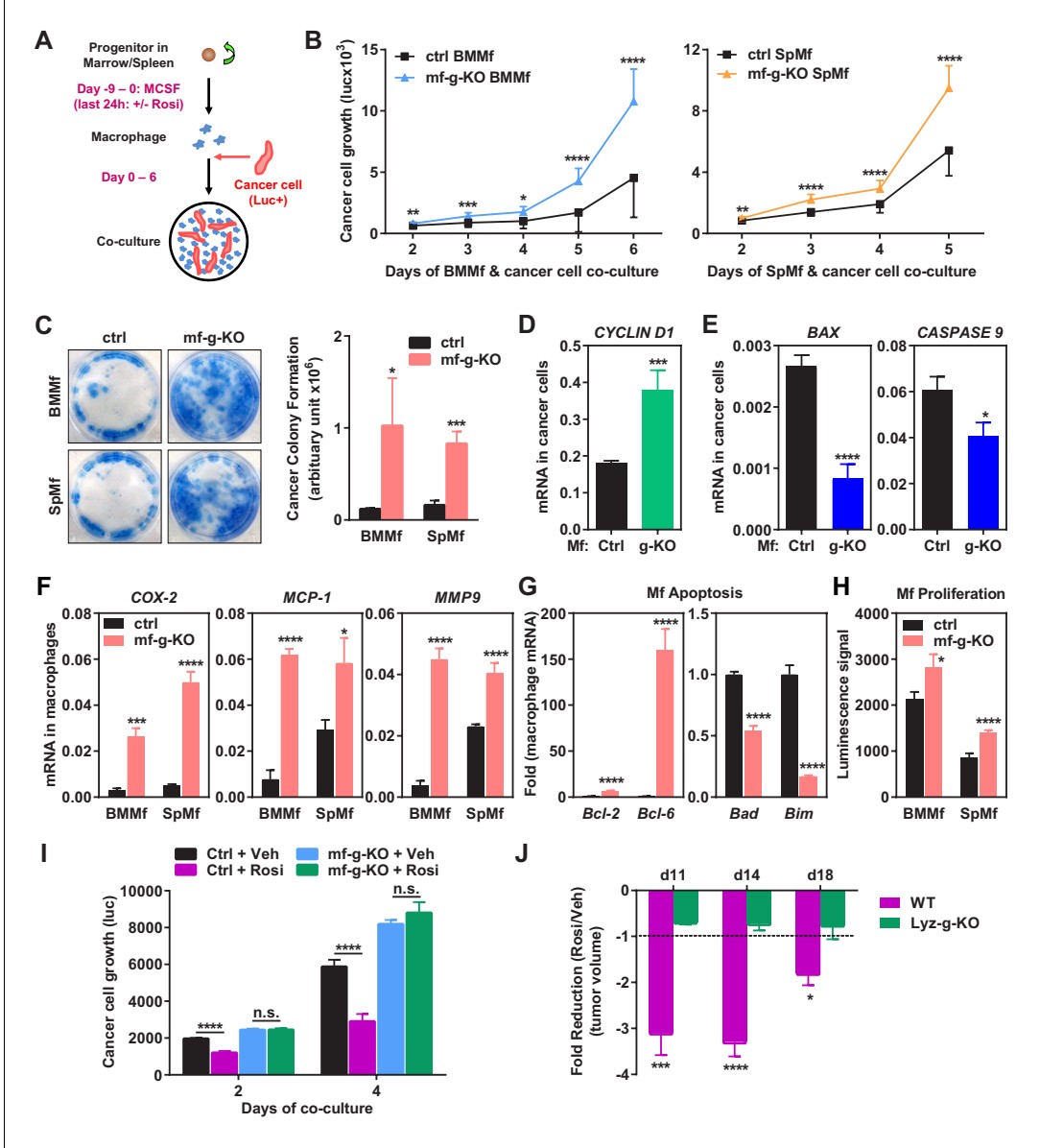

**Figure 2.** Macrophage PPARγ deletion exacerbates breast cancer cell proliferation and attenuates the anti-tumor effect of rosiglitazone. (A) A diagram of mouse macrophage and human breast cancer cell co-culture. Progenitors in bone marrow or spleen were differentiated into macrophages with M-CSF for nine days before the seeding of luciferase-labelled 1833 human breast cancer cells to the cultures. For rosiglitazone (Rosi) pre-treatment, macrophages were treated with 1 μM Rosi or vehicle control for the last 24 hr of macrophage differentiation; after medium was removed and cells were washed, cancer cells were added to the macrophage cultures in fresh medium without Rosi or vehicle. (B) Cancer cell proliferation was increased when co-cultured with PPARγ-deficient macrophages derived from bone marrow (left) or spleen (right) of mf-g-KO mice compared with WT control macrophages (n = 3). Cancer cell growth was quantified by luciferase signal for 2–6 days. (C) PPARγ-deficient macrophages promoted tumor cell colony formation in the co-cultures (n = 3). Tumor cells were cultured for 11–12 days for the colonies to form. Left, representative images of crystal violet staining. Right, quantification of colony formation. (D–E) Co-culture with PPARγ-deficient macrophages resulted in higher expression of proliferation markers (D) and lower expression of apoptosis markers (E) in breast cancer cells (n = 3). Human gene expression in cancer cells was quantified by RT-QPCR and human-specific primers. (F) PPARγ-deficient macrophages exhibited a higher expression of pro-inflammatory genes (n = 3). BMMf, bone marrow macrophage; SpMf, spleen macrophage. (G) PPARγ-deficient macrophages displayed higher levels of anti-apoptotic genes (left) and lower levels of pro-apoptotic genes (right) (n = 3). (H) PPARγ-deficient macrophages showed increased proliferation (n = 3). The number of metabolically active cells was determined by ATP content using the CellTiter-Glo Assay. (I) Co-culture with Rosi pre-treated macrophages inhibited breast cancer cell growth compared with vehicle (Veh) pre-treated macrophages in a macrophage-PPARγ-dependent manner (n = 3). (J) The ability of Rosi to suppress tumor growth in vivo was significantly attenuated in mf-g-KO mice (n = 6). Four days after EO771 cell mammary fat pad injection, mf-g-KO mice or control mice were treated with Veh or Rosi (10 mg/kg) every two days before tumor volume measurement. Error bars, SD; *p<0.05; **p<0.01; ***p<0.005; ****p<0.001; n.s. non-significant.

*Figure 2 continued on next page*

*Figure 2 continued*

The following figure supplement is available for figure 2:

**Figure supplement 1.** Additional analyses of co-cultures and rosiglitazone treatment.

searching published microarray databases comparing PPARγ-deficient vs. control macrophages (*Hevener et al., 2007*) and rosiglitazone- vs. vehicle-treated macrophages (*Welch et al., 2003*), we selected several candidate membrane proteins that might be regulated by PPARγ in macrophages. Upon examining their expression in our macrophage cultures, we found that G protein-coupled receptor 132 (Gpr132, also known as G2A) was consistently and significantly upregulated in PPARγ-deficient macrophages compared with WT control macrophages (see below), whereas the expression of 11 other candidates was unaltered (*Figure 3—figure supplement 1*). Therefore, we decided to further investigate whether Gpr132 is a functional PPARγ target gene in macrophages.

Gpr132 has been previously described as a stress-inducible seven-pass transmembrane receptor that functions at the G2/M checkpoint of the cell cycle (*Weng et al., 1998*), which modulates immune cell function (*Kabarowski, 2009*; *Radu et al., 2004*; *Yang et al., 2005*). We found that Gpr132 was predominantly expressed in the hematopoietic cell types/tissues and highly expressed in macrophages, but largely absent in other tissues or tumor cells (*Figure 3C–D*), indicating that it may play an important role in macrophage function. *Gpr132* expression was significantly higher in PPARγ-deficient macrophages compared with control macrophages, either in macrophage cultures alone or in macrophages co-cultured with cancer cells (*Figure 3E–F*). In line with this observation, PPARγ activation by rosiglitazone reduced Gpr132 expression in WT macrophages but not PPARγ-deficient macrophages (*Figure 3G*). These findings suggest that PPARγ represses *Gpr132* expression.

## PPARγ binds to *Gpr132* promoter and represses its transcriptional activity

To determine whether *Gpr132* is a direct PPARγ transcriptional target, we investigated if PPARγ can bind to the *Gpr132* promoter and regulate its transcription. *Gpr132* promoter regions (0.5 kb and 1 kb) were cloned into a luciferase reporter vector. Transient transfection and reporter assays reveal that luciferase output from both 0.5 Kb and 1 Kb *Gpr132* promoter was reduced by the co-transfection of PPARγ and further diminished by rosiglitazone treatment (*Figure 3H*). These results indicate that PPARγ represses *Gpr132* promoter via critical element(s) within the 500 base pairs upstream of *Gpr132* transcription start site. Indeed, we identified a PPAR response element (PPRE) half site in this region (-188: CATCCGAGCA**AGGTCA**GAC). Chromatin-immunoprecipitation (ChIP) assay showed that PPARγ could bind to the endogenous *Gpr132* proximal promoter in macrophages but not an upstream negative control region (*Figure 3I*); this binding was not significantly altered by rosiglitazone (not shown). Moreover, ChIP assay revealed that H3K9Ac active transcription histone mark at the Gpr132 transcriptional start site was enhanced upon PPARγ knockdown and reduced by rosiglitazone in a PPARγ-dependent manner (*Figure 3J*). These mechanistic studies indicate that PPARγ directly represses *Gpr132* transcription in macrophages upon activation by either endogenous or synthetic ligands.

## Gpr132 is repressed by PPARγ in human macrophages and correlates with human breast cancer

*Gpr132* expression in human macrophages derived from human peripheral blood mononuclear cells (hPBMN) was also blunted by rosiglitazone (*Figure 4A*). This indicates that the PPARγ repression of *Gpr132* is evolutionarily conserved and our findings in mice may translate to human physiology and disease. To explore the significance of Gpr132 in human breast cancer, we analyzed the RNA-Seq and clinical data of breast invasive carcinoma (BRCA) from The Cancer Genome Atlas (TCGA) database. Because *Gpr132* is highly expressed in human macrophages (*Figure 4A*) but absent in human breast cancer cells (*Figure 3D*), the *Gpr132* expression in tumors mainly originates from hematopoietic cells in the microenvironment such as macrophages. Compared with normal breast samples, the

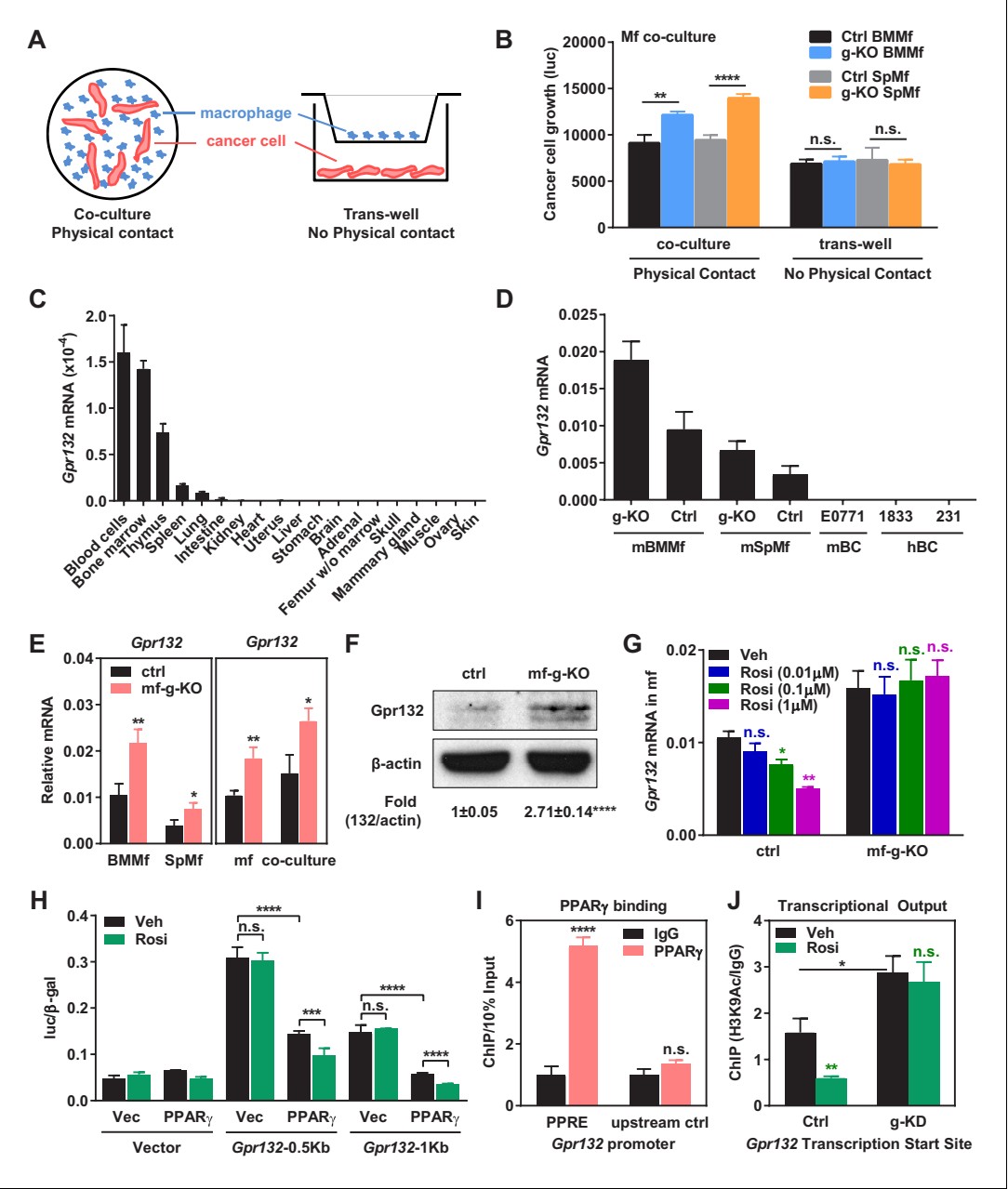

**Figure 3.** PPARγ represses Gpr132 transcription in macrophages. (A–B) Physical contact is required for the pro-tumor effects of PPARγ-deficient macrophages. (A) A schematic diagram of the co-culture vs. trans-well systems. (B) Tumor cell proliferation was enhanced by co-culture with PPARγ-deficient macrophages but not by their conditioned medium delivered via trans-well (n = 3). (C) Gpr132 was predominantly expressed in the hematopoietic cell types and tissues (n = 3). (D) Gpr132 was expressed in macrophages but largely absent in breast cancer cells. mBMMf, mouse bone marrow macrophage; mSpmf, mouse spleen macrophage; mBC, mouse breast cancer cells; hBC, human breast cancer cells. (E) Gpr132 mRNA levels were significantly higher in PPARγ-deficient macrophages compared with control macrophages, either in macrophage cultures alone or in macrophages co-cultured with human breast cancer cells (n = 3). (F) Gpr132 protein expression was significantly higher in PPARγ-deficient macrophages (n = 3). (G) PPARγ activation by rosiglitazone reduced Gpr132 mRNA in WT macrophages but not in PPARγ-deficient macrophages (n = 3). (H) Transcriptional output from both 0.5 Kb and 1 Kb Gpr132 promoters was reduced by the co-transfection of PPARγ and further diminished by rosiglitazone (n = 3). HEK293 cells were transfected with PPARγ and its heterodimer partner retinoic X receptor α (RXRα), together with a luciferase reporter driven by 0.5 Kb or 1 Kb Gpr132 promoter, and compared with vector-transfected controls. Next day, cells were treated with rosiglitazone or vehicle control for 24 hr before harvest and

*Figure 3 continued on next page*

*Figure 3 continued*

reporter analyses. (I) ChIP assay of PPARγ binding to the endogenous Gpr132 promoter in macrophages. A PPRE region in the Gpr132 promoter was pulled down with anti-PPARγ antibody or an IgG control antibody in RAW264.7 mouse macrophages and detected by QPCR (n = 3). An upstream Gpr132 promoter region served as a negative control. (J) ChIP assay of H3K9Ac active transcription histone mark at the Gpr132 transcription start site showed that rosiglitazone represses the transcriptional activity from a Gpr132 promoter in a PPARγ-dependent manner (n = 3). Control or PPARγ knockdown (KD) RAW264.7 macrophages were treated with 1 µM rosi or vehicle control for 4 hr before harvest. Error bars, SD; *p<0.05; **p<0.01; ***p<0.005; ****p<0.001; n.s. non-significant.

The following figure supplement is available for figure 3:

**Figure supplement 1.** Expression of other candidate genes was unaltered in PPARγ-deficient macrophages.

majority of breast cancer lesions displayed significantly higher *Gpr132* expression (*Figure 4B*); in addition, compared with ER-positive breast cancers, the more aggressive ER-negative breast cancers also exhibited higher *Gpr132* expression (*Figure 4C*). Immunohistochemistry staining confirmed that human breast cancer tissues expressed significantly higher *Gpr132* compared with normal breast control tissues (*Figure 4D–E*). Moreover, linear regression analyses showed that higher *Gpr132* expression was significantly correlated with higher expression of pro-inflammatory markers including *CCL2 (MCP-1), MMP9* and *PTGS2 (COX-2)* in breast cancer lesions (*Figure 4F*). These findings further suggest that macrophage Gpr132 may promote inflammation and tumor progression.

## Macrophage Gpr132 facilitates cancer cell proliferation in vitro

We next examined the function of macrophage Gpr132 in regulating cancer cells using our in vitro co-culture system. Gpr132 knockdown in macrophages significantly reduced cancer cell growth (*Figure 5A–C*). Conversely, Gpr132 over-expression in macrophages increased cancer cell growth (*Figure 5D–F*). The anti-Gpr132 antibody was validated using Gpr132-KO mice (*Figure 5—figure supplement 1A*). We then compared macrophages derived from the bone marrow or spleen of Gpr132-KO mice vs. littermate WT control mice. Gene expression analyses reveal that Gpr132-KO macrophages displayed the opposite phenotype from PPARγ-deficient macrophages, with lower pro-inflammatory genes (*Figure 5G*), higher pro-apoptotic genes and lower anti-apoptotic genes (*Figure 5H*) compared with WT macrophages. In vitro macrophage-tumor cell co-culture experiments showed that Gpr132-KO macrophages exhibited a significantly reduced ability to promote cancer cell colony formation and growth (*Figure 5I–J*). These results indicate that Gpr132 enhances inflammation and macrophage survival, and the upregulated Gpr132 in PPARγ-deficient macrophages may confer their tumor-promoting effects.

## Gpr132 knockout mice support less tumor growth in vivo

To examine the effects of Gpr132 deletion in the tumor environment on cancer growth in vivo, we injected EO771 mouse breast cancer cells into the mammary fat pad of Gpr132-KO mice and WT littermate controls. In this system, Gpr132 was deleted in macrophages as well as other Gpr132-expressing tissues such as bone marrow, spleen and thymus, but not in the injected cancer cells which had essentially no Gpr132 expression (*Figure 3C–D*). Previous study show that Gpr132-KO mice display a normal pattern of T and B lineage differentiation, appearing healthy and indistinguishable from WT littermates throughout young adulthood, but develop progressive secondary lymphoid organ enlargement associated with abnormal expansion of both T and B lymphocytes that become pathological when older than one year of age (*Le et al., 2001*). Therefore, our experiments were initiated in young mice and terminated before Gpr132-KO mice aged to prevent any potential effects of lymphoid defects on cancer growth. Compared with WT and Gpr132 heterozygous (Het) controls, Gpr132-KO mice exhibited significantly diminished tumor growth (*Figure 5K*). Compared with WT controls, Gpr132-Het mice also showed attenuated tumor growth at a later stage (*Figure 5K*), indicating that Gpr132 regulation is dosage-sensitive. Together, our in vitro and in vivo results indicate that macrophage Gpr132 promotes tumor growth, suggesting that Gpr132 inhibition may impede cancer progression.

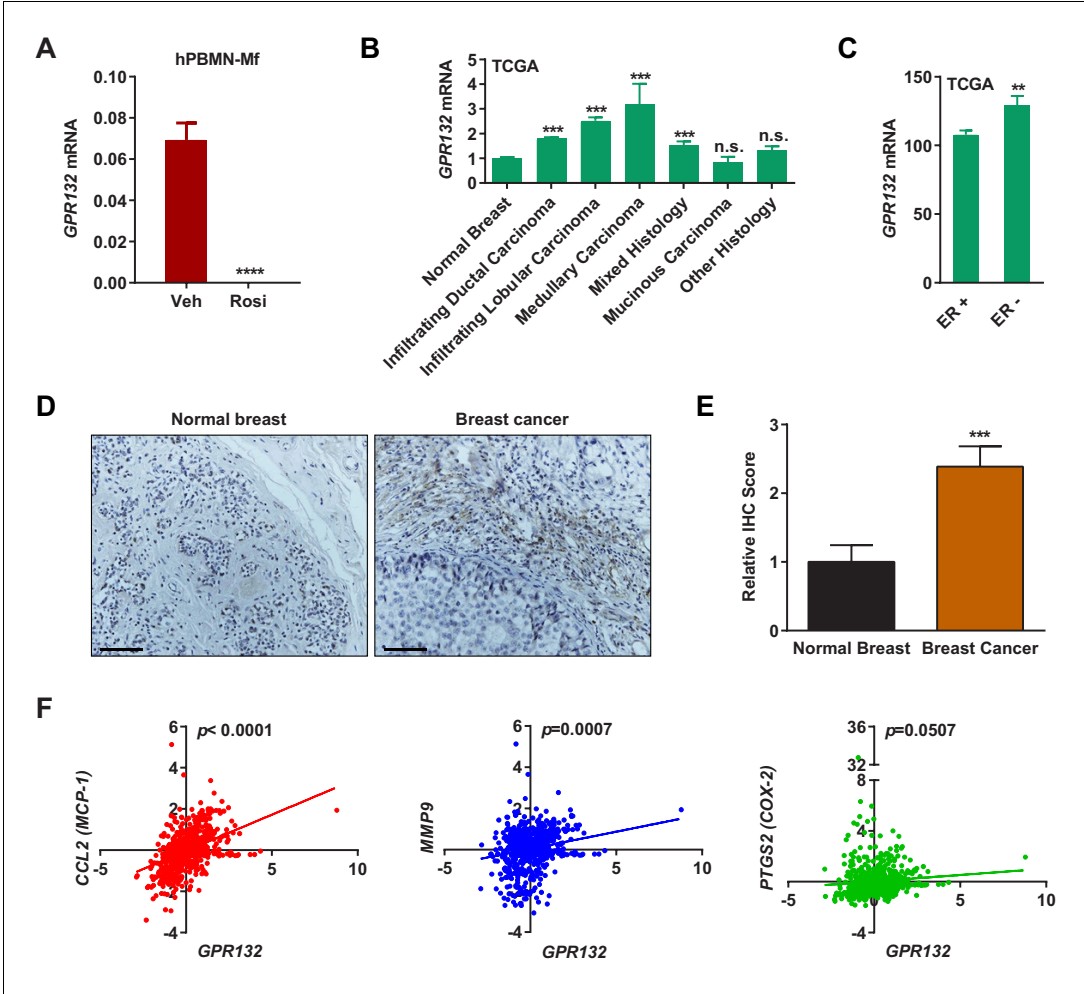

**Figure 4.** Gpr132 is repressed by PPARγ in human macrophages and correlates with human breast cancer. (**A**) Human Gpr132 expression in hPBMN-derived macrophages was blunted by rosiglitazone treatment (n = 3). Macrophages were treated with 1 µM rosiglitazone or vehicle for 4 hr. (**B**) TCGA BRCA data analysis showed that compared with normal breast samples, breast cancer lesions displayed higher Gpr132 expression. Normal Breast (n = 111); Infiltrating Ductal Carcinoma (n = 750); Infiltrating Lobular Carcinoma (n = 168); Medullary Carcinoma (n = 5); Mixed Histology (n = 29); Mucinous Carcinoma (n = 14); Other Histology (n = 44). Error bars, SE. (**C**) TCGA BRCA data analysis showed that compared with $ER^+$ breast cancers (n = 746), $ER^-$ breast cancers (n = 221) exhibited higher Gpr132 expression. Error bars, SE. (**D–E**) Immunohistochemistry (IHC) of human tissue microarrays showed higher Gpr132 expression in breast cancer tissues (n = 16) compared with normal breast tissues (n = 13). Tissues were stained with anti-Gpr132 (brown) and hematoxylin (blue). (**D**) Representative images. Scale bars, 200 µm. (**E**) Quantification of relative IHC scores. Error bars, SE. (**F**) Linear regression analyses of TCGA BRCA data showed that Gpr132 expression was positively correlated with the expression of CCL2 (MCP-1), MMP9 and PTGS2 (COX-2) in breast cancer lesions (n = 805). Error bars, SD (**A**) or SE (**B,C,E**); *$p<0.05$; **$p<0.01$; ***$p<0.005$; ****$p<0.001$; n.s. non-significant.

## Macrophage Gpr132 mediates PPARγ regulation and rosiglitazone effects

To further examine whether Gpr132 is a functional PPARγ target in macrophage that is required for PPARγ cancer regulation, we conducted pharmacological and genetic experiments. As a pharmacological gain-of-function strategy, we treated the macrophage-cancer cell co-cultures or tumor-grafted mice with rosiglitazone or vehicle control. Pre-treating Gpr132-KO macrophages with rosiglitazone before cancer cell seeding no longer showed any inhibition of cancer cell proliferation in the co-cultures (*Figure 5L*) (*Figure 5—figure supplement 1B*). Consistent with this in vitro observation,

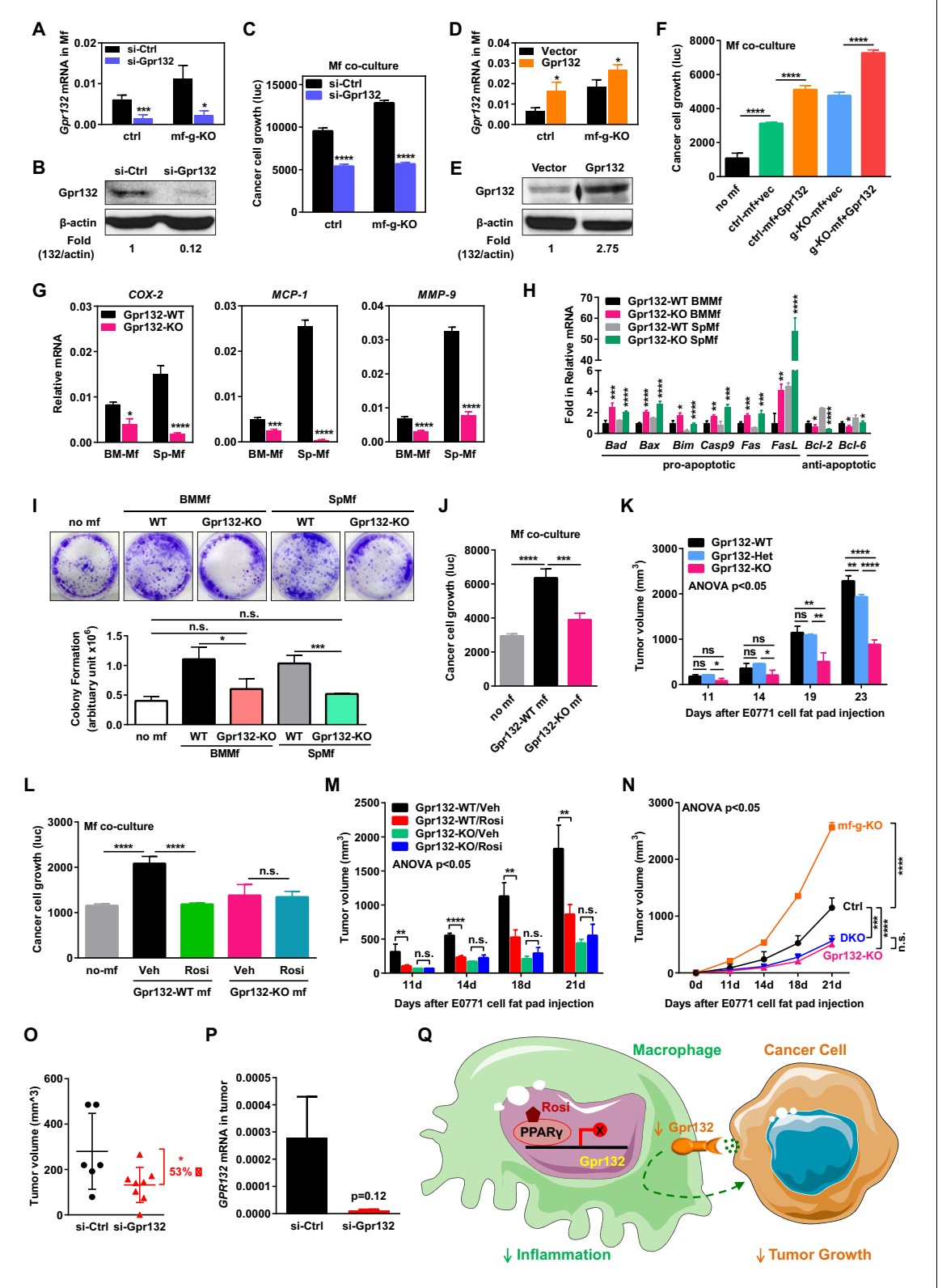

**Figure 5.** Macrophage Gpr132 promotes tumor growth and mediates the anti-tumor effect of rosiglitazone. (A–B) Gpr132 knockdown decreased Gpr132 mRNA (A) and protein (B) in macrophages (n = 3). (C) In co-cultures, Gpr132 knockdown in macrophages reduced cancer cell growth (n = 3). (D–E) Gpr132 over-expression increased both mRNA (D) and protein (E) in macrophages (n = 3). (F) In co-cultures, Gpr132 over-expression in macrophages enhanced cancer cell growth (n = 3). Cancer cell alone without macrophages (no mf) served as a negative control. (G) Gpr132-KO

*Figure 5 continued on next page*

*Figure 5 continued*

macrophages exhibited lower expression of pro-inflammatory genes compared with WT controls (n = 3). (H) Gpr132-KO macrophages displayed higher levels of pro-apoptotic genes and lower levels of anti-apoptotic genes (n = 3). (I–J) In vitro co-cultures showed that Gpr132 deletion in macrophages significantly reduced the ability of macrophages to promote cancer cell colony formation (I) and proliferation (J) (n = 3). (K) In vivo mammary fat pad tumor grafts showed that tumor growth was significantly diminished in Gpr132-KO mice compared with WT or Gpr132-Het mice (n = 6). (L) In in vitro co-cultures, Rosi pre-treated WT macrophages but not Rosi pre-treated Gpr132-KO macrophages were able to inhibit cancer cell growth (n = 3). (M) The ability of Rosi to suppress tumor growth in vivo was abolished in Gpr132-KO mice (n = 6). Four days after EO771 cell mammary fat pad injection, Gpr132-KO or WT mice were treated with Veh or Rosi (10 mg/kg) every two days. (N) The ability of macrophage PPARγ deletion to exacerbate tumor growth in vivo was abolished in Gpr132-KO mice (n = 4). DKO, mf-g/Gpr132 double KO. (O–P) Pharmacological Gpr132 inhibition impeded mammary tumor growth. Female mice (six-week-old) were treated with si-Gpr132 (n = 8) or si-Ctrl (n = 6) for 18 days via intravenous injection at 10 µg/mouse twice/week, three days before and 15 days after EO771 cell mammary fat pad injection. (O) Tumor volume was significantly decreased by si-Gpr132 treatment. (P) Gpr132 expression in tumors was effectively depleted. Error bars, SD; *p<0.05; **p<0.01; ***p<0.005; ****p<0.001; n.s. non-significant. (Q) A simplified working model for how macrophage PPARγ inhibits inflammation and tumor growth by repressing the transcription of macrophage Gpr132, a novel pro-inflammatory and pro-tumor membrane receptor. Upon sensing and activation by tumor signals, macrophage Gpr132 may modulate macrophage intracellular signaling and downstream targets, which in turn promotes cancer cell proliferation (indicated by the dashed line). Macrophage PPARγ deficiency increases Gpr132 level to afford better tumor sensing by macrophages, thereby promoting tumor growth. Pharmacological Gpr132 inhibition via either PPARγ agonist or Gpr132 blockade attenuates breast cancer progression. Moreover, both macrophage PPARγ and Gpr132 are key mediators of the anti-tumor effects of the clinically used TZD drug rosiglitazone.

The following figure supplement is available for figure 5:

**Figure supplement 1.** Additional analysis of Gpr132.

the anti-tumor effect of rosiglitazone in vivo was also abolished in Gpr132-KO mice (*Figure 5M*). These pharmacological findings support that the PPARγ repression of Gpr132 in macrophages is a significant contributor to the anti-tumor effects of rosiglitazone.

As a genetic loss-of-function strategy, we bred Gpr132-KO mice with mf-g-KO mice to generate mf-g/Gpr132 double KO (DKO) mice. Mammary fat pad tumor graft experiments demonstrated that Gpr132 deletion in the DKO mice impaired the ability of macrophage PPARγ deficiency to exacerbate tumor growth because DKO mice showed similar tumor volume as Gpr132-KO mice (*Figure 5N*). This genetic rescue further supports that Gpr132 is an essential mediator of macrophage PPARγ regulation of breast cancer progression.

## Pharmacological Gpr132 inhibition impedes tumor growth

To further explore Gpr132 as a potential cancer therapeutic target, we next examined whether acute pharmacological inhibition of Gpr132 could attenuate breast cancer progression. Because macrophage precursors reside in hematopoietic tissues such as blood, bone marrow and spleen that can be efficiently targeted by siRNA (*Larson et al., 2007*), we chose to employ siRNA-mediated Gpr132 knockdown. We treated WT female mice with si-Gpr132 or si-Ctrl for 18 days via intravenous injection at 10 µg/mouse twice/week, three days before and 15 days after cancer cell graft. The results showed that si-Gpr132 significantly reduced tumor volume compared with si-Ctrl (*Figure 5O*), as the result of depleted Gpr132 expression (*Figure 5P*) (*Figure 5—figure supplement 1C*). Body weight was unaltered by si-Gpr132 (not shown), indicating a lack of overt toxicity by Gpr132 inhibition. These results support the Gpr132 inhibition as a novel anti-cancer strategy.

## Discussion

Given the pleotropic and important roles of PPARγ in physiology and disease, as well as the widespread usage of TZD drugs for the treatment of insulin resistance and type II diabetes, it is of paramount importance to elucidate the mechanisms for how PPARγ and TZDs affect cancer. Here we have uncovered a crucial yet previously unrecognized role of macrophage PPARγ in suppressing cancer progression and mediating the anti-tumor effects of rosiglitazone (*Figure 5Q*). Mechanistically, PPARγ activation in macrophages tunes down inflammatory programs by repressing the transcription of a novel target gene Gpr132, which is a pro-inflammatory membrane receptor (*Figure 5Q*). Consequently, tumor growth is inhibited when the macrophage Gpr132 level is low by either Gpr132 deletion/inhibition or PPARγ activation via rosiglitazone; whereas tumor growth is exacerbated when

the macrophage Gpr132 level is high as the result of macrophage PPARγ deficiency. Importantly, Gpr132 deletion abolishes the cancer regulation by macrophage PPARγ or rosiglitazone, indicating that Gpr132 is an essential mediator of PPARγ functions in macrophages and tumor progression. These findings reveal PPARγ and Gpr132 as fundamental key players in TAM, providing new mechanisms how macrophages interact with tumor cells to promote cancer malignancy.

Although our co-culture experiments suggest that tumor cell-macrophage contact is required for macrophage PPARγ regulations, other possible mechanisms may exist. For example, (1) tumor cell-macrophage close proximity (rather than direct contact) may be required so that an increased macrophage Gpr132 expression can better sense a gradient of the signal sent from tumor cells - this is supported by previous studies indicating Gpr132 as a pH and lipid sensor that may regulate immune cell trafficking (*Justus et al., 2013*; *Vangaveti et al., 2010*); (2) upon activation by tumor signals, macrophage Gpr132 may modulate macrophage intracellular signaling and downstream targets, which in turn promotes cancer cell proliferation (*Figure 5Q*). Our findings have opened an exciting new path for future investigations to delineate how macrophage Gpr132 senses tumor signals and then exacerbates tumor malignancy.

Both inflammatory macrophages (M1-like) and M2-like macrophages have been shown to be potentially pro-tumor. Recent literature strongly support that inflammation exacerbates tumor progression, as highlighted in several reviews (*Coussens and Werb, 2002*; *Noy and Pollard, 2014*; *Qian and Pollard, 2010*). Thus, in terms of how macrophage PPARγ impacts tumor progression, only the phenotype speaks of the truth. Our in vivo and in vitro findings all support an anti-inflammatory and anti-tumor role of macrophage PPARγ activation, and a pro-inflammatory and pro-tumor effect of macrophage PPARγ deletion. Among the many downstream events that mediate the pro-tumor effects of inflammatory macrophages, it is possible that macrophage PPARγ not only regulates cancer cell behavior but also modulates tumor-infiltrating lymphocytes (TILs) to alter immune surveillance (*Engblom et al., 2016*).

PPARγ may act on several cell types to exert anti-tumor effects including previously described cancer cells (*Mueller et al., 1998*; *Srivastava et al., 2014*; *Tontonoz et al., 1997*) and now reported immune cells. Our findings reveal macrophage as a key cell type, if not the only cell type, that is essential for the tumor suppressive effects of TZDs. Interestingly, although Tie2-g-KO mice harbored more complete PPARγ deletion than Lyz-g-KO mice and both mouse models showed enhanced tumor growth and tumor macrophage recruitment, the phenotype was more pronounced in Lyz-g-KO. It is possible that the PPARγ deletion in other hematopoietic cells outside of the myeloid lineage exerted opposite albeit minor effect on tumor growth in the Tie2-g-KO mice, which was absent in the Lyz-g-KO mice.

Similarly, PPARγ may affect many genes in macrophage, and targets other than Gpr132 may also contribute to the anti-tumor effects of PPARγ. Nonetheless, our genetic rescue and pharmacological experiments indicate that Gpr132 is a very important part of the puzzle and an essential mediator of macrophage PPARγ regulation, because in the absence of Gpr132, the tumor-modulating function of macrophage PPARγ or rosiglitazone is abolished. This uncovers Gpr132 as not only a novel key PPARγ target but also a new cancer therapeutic target.

Our luciferase reporter assays indicate that PPARγ directly suppresses the transcriptional activity from Gpr132 promoter, which is further supported by ChIP assays showing PPARγ binding to Gpr132 promoter, leading to a decreased level of H3K9Ac active transcription histone mark at the Gpr132 transcriptional start site. Nonetheless, additional mechanisms such as changes in mRNA stability may also contribute to PPARγ down-regulation of Gpr132 mRNA level.

Cancer cells form an intimate relationship with TAMs to proliferate and survive. As such, targeting the infiltrating macrophages to alter their number and properties can lead to a significant inhibition of cancer malignancy. Our data suggest that this can be achieved by either PPARγ activation or Gpr132 inhibition in the macrophage. By elucidating the mechanisms that macrophages use to promote cancer and inflammation, effective diagnostic tools as well as innovative anti-tumor and anti-inflammatory therapeutics can be designed. For example, macrophage levels of PPARγ and Gpr132 may predict not only tumor aggressiveness but also the pharmacological responses to rosiglitazone or Gpr132 inhibitors. Our findings may explain why rosiglitazone exerts anti-tumor effects in certain cancers but not others – cancers with abundant PPARγ-positive macrophages may be sensitive whereas cancers with limited macrophages or PPARγ-negative macrophages may be resistant.

Remarkably, the positive association of Gpr132 with inflammation and breast cancer in human (*Figure 4B–F*), the repression of Gpr132 expression by rosiglitazone in human macrophage (*Figure 4A*) and the anti-tumor effects of pharmacological Gpr132 inhibition (*Figure 5O–P*) highlight the exciting potential of Gpr132 blockade as a new therapeutic. Moreover, the observation that Gpr132 expression is significantly increased in the majority of human breast cancers (*Figure 4B*) suggests that Gpr132 may serve as a useful marker for breast cancer prognosis, similar to the 21 genes typically examined in the commercially available Oncotype Dx panel.

In summary, the significance of our findings resides in the following aspects: (1) it reveals macrophage as an important cell type that contributes to PPARγ suppression of cancer and the anti-tumor effects of rosiglitazone; (2) it identifies Gpr132 as a novel PPARγ direct target gene in macrophages that mediates PPARγ functions; (3) it uncovers Gpr132 as a pro-inflammatory and pro-tumor factor in macrophages, and thus a novel therapeutic target. Ultimately, these new knowledge will enhance our understanding of macrophage regulation, cancer microenvironment as well as PPARγ and Gpr132 biology, which may translate to a better intervention of diseases such as cancer, diabetes and inflammatory disorders.

## Materials and methods

### Mice

PPARγ flox mice on a C57BL/6J background (RRID:IMSR_JAX:004584) were described (*He et al., 2003*). Gpr132 knockout mice on a C57BL/6J background (RRID:IMSR_JAX:008576) (*Le et al., 2001*) were obtained from the Jackson Laboratory. Mice were fed standard chow ad libitum and kept on a 12-h light, 12-h dark cycle. PPARγ flox mice were bred with Tie2-Cre (RRID:IMSR_JAX:008863) (*Kisanuki et al., 2001*) or Lysozyme-Cre (RRID:IMSR_JAX:004781) (*Clausen et al., 1999*) transgenic mice to generate mf-g-KO mice. Tie2-g-KO was bred with Gpr132-KO to obtain mf-g/Gpr132 double KO mice. Representative results for Tie2-g-KO are shown unless specified as Lyz-g-KO. All experiments were conducted using littermates. Sample size estimate was based on power analyses performed using SAS 9.3 TS X64_7PRO platform. All protocols for mouse experiments were approved under Animal Protocol Number 2008–0324 by the Institutional Animal Care and Use Committee of UTSW.

### Macrophage and cancer cell cultures

For bone marrow- and spleen-derived macrophage, mouse bone marrow or splenocyte were collected with serum-free DMEM. After passing through a 40 μm cell strainer, the cells were cultured in macrophage differentiation medium (DMEM + 10% FBS + 20 ng/ml M-CSF) for six days. Gpr132 overexpression was performed with lentiviral transduction. The RAW264.7 mouse macrophage cell line (RRID:CVCL_0493) was from ATCC. The EO771 cell line originally derived from a spontaneous mammary tumor in a C57BL/6 mouse (RRID: CVCL_GR23) (*Casey et al., 1951*) was from CH3 BioSystems (Amherst, NY). The luciferase-labeled MDA-MB-231 human breast cancer sub-line (MDA-BoM-1833; RRID:CVCL_DP48) (*Kang et al., 2003*) was provided by Joan Massagué (Memorial Sloan-Kettering Cancer Center). The luciferase-labeled 4T1.2 mouse mammary tumor subline (RRID:CVCL_GR32) (*Lelekakis et al., 1999*) was provided by Robin Anderson (Peter MacCallum Cancer Centre) and Yibing Kang (Princeton University). Cell lines were authenticated by STR profiling and verified negative for mycoplasma. For macrophage and cancer cell co-cultures, mouse bone marrow and spleen cells were plated in 96-well plate and differentiated into macrophages with 20 ng/ml M-CSF for nine days. Luciferase-labeled 1833 cells or 4T1.2 cells were then added to the culture dish. At the end point of experiment, cell lysates were collected for luciferase assay to assess cancer cell growth. For the pre-treatment, macrophages were cultured with 1 μM rosiglitazone (Cayman Chemical, Ann Arbor, MI) for the last 24 hr; the medium was removed and the macrophages were washed before cancer cell seeding.

### Orthotopic fat pad injection of mouse breast cancer cells

EO771 cells ($2.5 \times 10^5$ or $5 \times 10^5$) were injected into the mammary fat pad of 6–8 weeks old female mice. EO771 cells were prepared with a 1:1 ratio in the blank RPMI-1640 medium and matrigel (BD Biosciences, San Jose, CA). Every 2–3 days, tumor length and width were measured with a caliper

and tumor volume was calculated using the formula V = (L × W × W) / 2, where V is tumor volume, L is tumor length, and W is tumor width. The Py230 cell line was derived from spontaneous mammary tumors in C57BL/6 MMTV-PyMT female transgenic mice (RRID:CVCL_AQ08) (*Biswas et al., 2014*).

## Immunofluorescence staining

Tumor tissues were isolated from tumor-bearing mice three weeks after cancer cell injection. Tumors were frozen in OCT compound (Tissue-Tek), cryo-sectioned, and fixed with acetone before staining with antibodies. The tumor sections were blocked with 2% BSA, and then incubated with FITC anti-CD11b antibody (BD Pharmingen; RRID:AB_394774; 1:50 dilution) or FITC anti-F4/80 antibody (AbD Serotec, Raleigh, NC; RRID:AB_1102553; 1:50 dilution). For antibodies without FITC conjugate, the tumor sections were incubated with rat monoclonal anti-endomucin (Santa Cruz Biotechnologies, Dallas, TX; RRID:AB_2100037; 1:50 dilution), rabbit monoclonal anti-Ki67 (Cell Signaling, Danvers, MA; RRID:AB_2620142; 1:400 dilution), or rabbit polyclonal anti-Phospho-Histone H3 (Ser10) (Cell Signaling; RRID:AB_331534; 1:200 dilution). After washing with PBS, the sections were incubated with goat-anti-rat IgG-FITC antibody (RRID:AB_631753) or goat-anti-rabbit IgG-FITC antibody (RRID: AB_631744) (Santa Cruz Biotechnologies; 1:100 dilutions) for detection. After washing with PBS, cover slips were mounted with the Vectashield medium containing DAPI (Vector Laboratories, Burlingame, CA).

## Gene expression analyses

Tissue samples were snap frozen in liquid nitrogen and stored at −80°C. RNA was extracted using Trizol (Invitrogen, Carlsbad, CA) according to the manufacturer's protocol. RNA was first treated with RNase-free DNase I using the DNA-free kit (Ambion, Austin, TX) to remove all genomic DNA, and then reverse-transcribed into cDNA using an ABI High Capacity cDNA RT Kit (Invitrogen). The cDNA was analyzed using real-time quantitative PCR (SYBR Green, Invitrogen) with an Applied Biosystems 7700 Sequence Detection System. Each reaction was performed in triplicate in a 384-well format. The expression of mouse gene was normalized by mouse L19. The expression of the human gene was normalized with human GAPDH. Anti-Gpr132 antibody (Sigma, St. Louis, MO; RRID:AB_10745673) was validated using Gpr132-KO cells and used for western blot detection of Gpr132 protein.

## Immunohistochemistry

Tissue microarrays were purchased from US Biomax, Inc. (Rockville, MD), which contain human normal breast tissues and breast cancer tissues. The immunohistochemistry (IHC) staining was performed as previously described (*Su et al., 2014*; *Zhou et al., 2014*). Briefly, after dewaxing with xylence and dehydration with gradient ethanol, the tissue microarrays were incubated with antigen retrieval buffer (BD Biosciences) for 1 hr at 95°C, followed by treatment with 3% hydrogen peroxide (Sigma) for 10 min. Specimens were blocked with 5% defatted milk for 1 hr at room temperature, and incubated with anti-human-Gpr132 antibodies (Sigma; RRID:AB_10745673; 1:100) overnight at 4°C and then with HRP-conjugated secondary antibodies (1:200) for 30 min at room temperature. Immunostaining was performed using a diaminobenzidine (DAB) kit (Thermo scientific, Waltham, MA). The expression levels of Gpr132 were scored in a blind fashion semi-quantitatively according to the staining intensity and distribution using the immunoreactive score as described previously (*Su et al., 2014*; *Zhou et al., 2014*). Briefly, the IHC score = staining intensity (negative = 0; weak = 1; moderate = 2; and strong = 3) × percentage of positive cells (0% = 0; 0–25% = 1; 25–50% = 3; and 75–100% = 4).

## TCGA data analysis

RNA-Seq and clinical data of breast invasive carcinoma (BRCA) were downloaded from The Cancer Genome Atlas (TCGA) data portal (*Cancer Genome Atlas Network, 2012*) and tested for associations. Gene expression for GPR132, CCL2 (MCP-1), MMP9 and PTGS2 (COX-2) were analyzed by linear regression.

## Statistical analyses

All statistical analyses were performed with Student's t-Test and represented as mean ± standard deviation (SD) unless noted otherwise. For in vivo experiments with ≥3 groups, statistical analyses were performed with ANOVA followed by the post hoc Tukey pairwise comparisons. The p values were designated as *$p<0.05$; **$p<0.01$; ***$p<0.005$; ****$p<0.001$; n.s. non-significant ($p>0.05$).

## Acknowledgements

We thank Drs. Rolf Brekken, John Minna and Gray Pearson, as well as members of the Wan Laboratory for suggestions and discussion; Shengjun Fan for assistance with bioinformatics analyses. Y Wan is a Virginia Murchison Linthicum Scholar in Medical Research. This work was in part supported by CPRIT (RP130145, YW), DOD (W81XWH-13-1-0318, YW), NIH (R01DK089113, YW), Mary Kay Foundation (#073.14, YW), Simmons Cancer Center (YW), March of Dimes (#6-FY13-137, YW), The Welch Foundation (I-1751, YW), UT Southwestern Endowed Scholar Startup Fund (YW) and NCI Cancer Center Support Grant (5P30CA142543). The authors declare that they have no financial conflict of interest.

## Additional information

### Funding

| Funder | Grant reference number | Author |
|---|---|---|
| Cancer Prevention and Research Institute of Texas | RP130145 | Yihong Wan |
| U.S. Department of Defense | W81XWH-13-1-0318 | Yihong Wan |
| National Institute of Diabetes and Digestive and Kidney Diseases | R01DK089113 | Yihong Wan |
| Mary Kay Foundation | 073.14 | Yihong Wan |
| March of Dimes Foundation | 6-FY13-137 | Yihong Wan |
| Welch Foundation | I-1751 | Yihong Wan |
| National Cancer Institute | 5P30CA142543 | Yihong Wan |

The funders had no role in study design, data collection and interpretation, or the decision to submit the work for publication.

### Author contributions

WYC, Conception and design, Acquisition of data, Analysis and interpretation of data, Drafting or revising the article; HDH, PC, SP-L, Acquisition of data, Analysis and interpretation of data, Contributed unpublished essential data or reagents; YW, Conception and design, Analysis and interpretation of data, Drafting or revising the article

### Author ORCIDs

Yihong Wan, http://orcid.org/0000-0003-0556-7017

### Ethics

Animal experimentation: All protocols for mouse experiments were approved under Animal Protocol Number 2008-0324 by the Institutional Animal Care and Use Committee of UT Southwestern Medical Center.

## Additional files

### Major datasets

The following previously published dataset was used:

| Author(s) | Year | Dataset title | Dataset URL | Database, license, and accessibility information |
|---|---|---|---|---|
| Cancer Genome Atlas Network | 2012 | Comprehensive molecular portraits of human breast tumours | http://cbioportal.org | The data can be explored via the ISB Regulome Explorer (http://explorer.cancerregulome.org/) and the cBio Cancer Genomics Portal (http://cbioportal.org) |

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
