## [Decision Letter]

Thank you for submitting your article "Macrophage PPARγ Inhibits Gpr132 to Mediate the Anti-Tumor Effects of Rosiglitazone" for consideration by *eLife*. Your article has been favorably evaluated by Charles Sawyers (Senior Editor) and three reviewers, one of whom is a member of our Board of Reviewing Editors.

The reviewers have discussed the reviews with one another and the Reviewing Editor has drafted this decision to help you prepare a revised submission.

Summary:

In this manuscript, the authors present the interesting observation that PPARg-dependent expression of Gpr132 in macrophages attenuates breast cancer growth using a co-culture system and in vivo experiments with hematopoietic cell-specific PPARg-deficient mice and Gpr132-deficient mice. Translational information was provided by data showing upregulation of Gpr132 and a positive correlation between Gpr132 and inflammatory markers in human breast cancer tissues.

The reviewers were in agreement that the work was potentially of interest to the general audience of *eLife*. All of the reviewers appreciated the large amount of work presented. At the same time, the review process identified several opportunities to strengthen the work.

Essential revisions:

1) The authors do not discuss previously published work from the Kittler laboratory which demonstrated that the anti-proliferative activity of pioglitazone manifests its antiproliferative activity in cancer cells (lung and breast) by reprogramming cell metabolism (Cell Metab 20, 650-661). It seems plausible that the effects observed could be explained by a shift in oxidative to glycolytic metabolism upon PPARγ knockdown. Also, recruitment of M1-polarized macrophages to the tumors in mice is likely to increase the number of TILs. This introduces the possibility that the effects may relate to decreased immune surveillance. This possibility is also not considered. The reviewers feel that it is essential that the authors provide a more balanced discussion of their results in the context of previously published work in this area.

2) Please specify which Cre driver was used for each experiment. Tie2-Cre expresses in endothelial cells, and therefore the possibility that PPARg in endothelial cells is contributing to tumor growth needs to be considered with this model. Have the authors done the tumor growth experiment in LysM-PPARg-/- Gpr132-/- mice?

3) The data in Figure 1 are compelling. However, the confirmatory data, in Py230 cells, is only presented as endpoint studies (presumably; not clearly explained). Tumor growth data over time together with the luciferase data would better support that the effect was confirmatory of that presented in Figure 1.

4) Similarly, full growth curve data for the tumor study should be presented in Figure 2. Further, the key in vitro experiment (Figure 2) would be bolstered by the inclusion of data showing that PPARγ target genes are induced.

5) Do E0771 and Py230 cells express PPARγ? How did the authors rule out direct effects of rosi on the cancer cells? Related to this point, for the in vitro studies presented in Figure 2 why do the investigators switch to a third cell line MDA231? Why were E0771 or Py230 cells not used? Was it merely to allow the study of mouse/human gene expression or did Rosi differentially effect these cells in vitro?

6) The in vivo effect of rosiglitazone on tumor progression should also be characterized in terms of macrophage infiltration in WT and mf-g-KO mice.

7) Tumor size differences were less pronounced in hematopoietic/endothelial PPARg-Tie2-cre than in PPARg-LysM-cre with EO771 tumor mammary cells. Is this due to differences in PPARg recombination in the two models? The authors should analyze the degree of PPARg deletion in purified tumor-cell infiltrating macrophages.

8) The authors show increased myeloid infiltration into tumors (based on immunofluorescence staining) in the setting of PPARg deficiency. Therefore, it is not surprising that inflammatory gene expression is enhanced since more leukocytes are present on site. Inflammatory gene expression should be analyzed within intra-tumor infiltrating macrophages (FACS sorted), therefore comparing the same number of cells obtained from WT and macrophage PPARg-deficient mice. FACS analysis of infiltrating populations should be more carefully analyzed. Such data is critical to support the conclusion that PPARg deletion changes the functional properties of tumor infiltrating macrophages.

9) There is little prior evidence for PPARg acting as a direct repressor. Has an effect on mRNA stability been considered? Regarding the data in Figure 3, what is the effect of rosiglitazone on PPARg binding to the Gpr132 promoter? ChIP assay with H3K9Ac to compare wild-type macrophages to PPARg deficient macrophages would be informative.

---

## [Author Response]

*The reviewers were in agreement that the work was potentially of interest to the general audience of eLife. All of the reviewers appreciated the large amount of work presented. At the same time, the review process identified several opportunities to strengthen the work.*

Essential revisions:

1) The authors do not discuss previously published work from the Kittler laboratory which demonstrated that the anti-proliferative activity of pioglitazone manifests its antiproliferative activity in cancer cells (lung and breast) by reprogramming cell metabolism (Cell Metab 20, 650-661). It seems plausible that the effects observed could be explained by a shift in oxidative to glycolytic metabolism upon PPARγ knockdown. Also, recruitment of M1-polarized macrophages to the tumors in mice is likely to increase the number of TILs. This introduces the possibility that the effects may relate to decreased immune surveillance. This possibility is also not considered. The reviewers feel that it is essential that the authors provide a more balanced discussion of their results in the context of previously published work in this area.

Thank you for the suggestions. We have now provided more extended and balanced discussion of these aspects. We have cited the Kittler paper describing how pioglitazone inhibits lung and breast cancer cell proliferation via a metabolic switch (Introduction, fourth paragraph). We have now also discussed that the downstream events of how inflammatory macrophage accumulation promotes cancer cell growth may involve tumor-infiltrating lymphocytes (TILs) and consequently altered immune surveillance (Discussion, third paragraph). To clarify, our PPARγ conditional KO mice harbor PPARγ deletion in hematopoietic/myeloid lineages, but not cancer cells, thus the effects observed were unlikely due to cancer cell metabolic changes. Our findings reveal macrophage as a key cell type, if not the only cell type, that is essential for the anti-cancer effects of TZDs (Discussion, fourth paragraph).

2) Please specify which Cre driver was used for each experiment. Tie2-Cre expresses in endothelial cells, and therefore the possibility that PPARg in endothelial cells is contributing to tumor growth needs to be considered with this model. Have the authors done the tumor growth experiment in LysM-PPARg-/- Gpr132-/- mice?

We have now specified the Cre driver in the Methods (first paragraph) and figure legend. We agree that Tie2Cre is less specific to myeloid lineage than LyzCre, yet Tie2Cre has the advantage of a more complete PPARγ deletion than LyzCre, as now clarified in the first paragraph of the Results. Our preliminary data show that Lyz-g/Gpr132-DKO also exhibited decreased tumor growth compared with Lyz-g-KO and WT controls. Even though angiogenesis was also slightly increased in the Tie2-g-KO tumors (Figure 1—figure supplement 1), the findings that both Tie2-g-KO and Lyz-g-KO show similar phenotype support the notion that macrophage is the major responsible cell type.

3) The data in Figure 1 are compelling. However, the confirmatory data, in Py230 cells, is only presented as endpoint studies (presumably; not clearly explained). Tumor growth data over time together with the luciferase data would better support that the effect was confirmatory of that presented in Figure 1.

We have now presented the time course for both bioluminescence and tumor volume (Figure 1—figure supplement 1).

*4) Similarly, full growth curve data for the tumor study should be presented in Figure 2. Further, the key* in vitro *experiment (Figure 2) would be bolstered by the inclusion of data showing that PPARγ target genes are induced.*

We have now presented the time course data in Figure 2. We have also included a positive control showing that rosiglitazone induces a previously reported PPARγ target gene LXRα in macrophages (Figure 2—figure supplement 1).

*5) Do E0771 and Py230 cells express PPARγ? How did the authors rule out direct effects of rosi on the cancer cells? Related to this point, for the* in vitro *studies presented in Figure 2 why do the investigators switch to a third cell line MDA231? Why were E0771 or Py230 cells not used? Was it merely to allow the study of mouse/human gene expression or did Rosi differentially effect these cells* in vitro*?*

PPARγ is expressed in all these breast cancer cell lines; however, rosi treatment was only performed on macrophages before cancer cell co-culture – the rosi-containing medium was removed and macrophages were washed before cancer cell seeding – thus there is no direct effect of rosi on cancer cells in our in vitro system. In our in vivo system, rosi may also exert direct effect on cancer cells, but our results showed this effect was not sufficient when PPARγ was deleted in macrophages, further supporting the essential role, if not the only role, of macrophage PPARγ in the TZD anti-tumor function. As explained in the first paragraph of the subsection “PPARγ-deficient macrophages promote cancer cell proliferation in vitro”, for our in vitro macrophage-cancer cell co-culture studies in Figure 2 we chose to use human luciferase-labelled MDA231 subline for two reasons: 1) the luciferase label in this breast cancer cell line distinguishes cancer cells from macrophages, thus permitting the specific quantification of cancer cell proliferation; 2) gene expression in cancer cells and macrophages can be distinguished by RT-QPCR using human-specific primers (for cancer cells) and mouse-specific primers (for macrophages).

*6) The* in vivo *effect of rosiglitazone on tumor progression should also be characterized in terms of macrophage infiltration in WT and mf-g-KO mice.*

Thank you for the suggestion. We have now included this data (Figure 2—figure supplement 1).

7) Tumor size differences were less pronounced in hematopoietic/endothelial PPARg-Tie2-cre than in PPARg-LysM-cre with EO771 tumor mammary cells. Is this due to differences in PPARg recombination in the two models? The authors should analyze the degree of PPARg deletion in purified tumor-cell infiltrating macrophages.

We have now clarified that macrophage PPARγ deletion was more complete in Tie2-g-KO compared with Lyz-g-KO (Results, first paragraph). Although both mf-g-KO mouse models showed enhanced tumor phenotype, as the reviewer pointed out, the phenotype was more pronounced in Lyz-g-KO. It is possible that PPARγ deletion in other hematopoietic cells outside of the myeloid lineage exerted opposite albeit minor effect on tumor growth in the Tie2-g-KO mice, which was absent in the Lyz-g-KO mice. We have now added this to Discussion (fourth paragraph).

8) The authors show increased myeloid infiltration into tumors (based on immunofluorescence staining) in the setting of PPARg deficiency. Therefore, it is not surprising that inflammatory gene expression is enhanced since more leukocytes are present on site. Inflammatory gene expression should be analyzed within intra-tumor infiltrating macrophages (FACS sorted), therefore comparing the same number of cells obtained from WT and macrophage PPARg-deficient mice. FACS analysis of infiltrating populations should be more carefully analyzed. Such data is critical to support the conclusion that PPARg deletion changes the functional properties of tumor infiltrating macrophages.

We concur with the reviewers that it is important to separate the effect of PPARγ deletion on macrophage abundance and macrophage properties. Although FACS sorting of tumor infiltrating macrophages has been used to examine macrophage gene expression, it is a common concern that this long procedure including 37°C collagenase digestion will alter gene expression. Instead, we tackled this question by employing a variety of systems to examine the intrinsic macrophage properties including bone marrow and spleen-derived macrophages, both in the absence and presence of breast cancer cell co-culture. All our results support a pro-inflammatory effect of PPARγ deletion and an anti-inflammatory function of rosiglitazone in macrophages.

9) There is little prior evidence for PPARg acting as a direct repressor. Has an effect on mRNA stability been considered? Regarding the data in Figure 3, what is the effect of rosiglitazone on PPARg binding to the Gpr132 promoter? ChIP assay with H3K9Ac to compare wild-type macrophages to PPARg deficient macrophages would be informative.

Our luciferase reporter assays indicate that PPARγ directly suppresses the transcriptional activity from Gpr132 promoter, which is further supported by ChIP assays showing PPARγ binding to Gpr132 promoter, leading to a decreased level of H3K9Ac active transcription histone mark at the Gpr132 transcriptional start site. Yet, additional mechanisms such as changes in mRNA stability may also contribute to the regulation. We have now discussed this point (Discussion, sixth paragraph). For our ChIP assays, we have clarified that rosiglitazone did not significantly alter PPARγ binding to Gpr132 promoter (subsection “PPARγ binds to Gpr132 promoter and represses its transcriptional activity”), and provided additional data showing that PPARγ-deficient macrophages displayed elevated H3K9Ac histone mark and resistance to rosi effects (Figure 3 and in the aforementioned subsection).